# Changes of Physicochemical Characteristics and Flavor during Suanyu Fermentation with *Lactiplantibacillus plantarum* and *Saccharomyces cerevisiae*

**DOI:** 10.3390/foods11244085

**Published:** 2022-12-17

**Authors:** Qiang Zhang, Naiyong Xiao, Huiya Xu, Zhihang Tian, Bowen Li, Weiqiang Qiu, Wenzheng Shi

**Affiliations:** 1College of Food Sciences & Technology, Shanghai Ocean University, Shanghai 201306, China; 2National R & D Branch Center for Freshwater Aquatic Products Processing Technology (Shanghai), Shanghai 201306, China

**Keywords:** *Lactiplantibacillus plantarum*, *Saccharomyces cerevisiae*, fermentation, microbial analysis, volatile compounds

## Abstract

This study investigates the changes of the physicochemical characteristics and flavor of fermented Suanyu (Chinese fermented fish) during fermentation with *Lactiplantibacillus plantarum* (*L. plantarum*) and *Saccharomyces cerevisiae* (*S. cerevisiae*). The related indicators, including pH, water activity (Aw), volatile base nitrogen (TVB-N), thiobarbituric acid (TBA), free amino acids (FAAs), microbial community, and volatile compounds were determined. *L. plantarum* fermentation samples (LP) and natural fermentation samples (NF) were used as controls. The pH and Aw of three groups of Suanyu samples decreased continuously through the entire fermentation process. Meanwhile, the TVB-N of three groups of samples increased gradually, while TBA first increased and then decreased. Notably, the pH, Aw, TVB-N, and TBA of MF group samples (inoculated *L. plantarum* and *S. cerevisiae*) were significantly lower than the NF group samples. In addition, both TVB-N and TBA of the MF group samples were lower than those of the LP group samples during fermentation, suggesting that combined fermentation could inhibit the growth of undesirable microorganisms more effectively. *Lactobacillus* were the main bacterial genus of the three group fermented samples during fermentation, and combined fermentation could promote the growth of *Lactobacillus* more significantly. In addition, the highest content of umami (145.16 mg/100 g), sweet amino acids (405.75 mg/100 g), and volatile compounds (especially alcohols and esters) were found in MF group samples, followed by the NF and LP group samples, indicating that combined fermentation could give Suanyu a better flavor quality. This study may provide a theoretical basis for the industrial production of fermented fish products and the improvement of fermentation technology.

## 1. Introduction

Suanyu, a representative fermented fish in China, is favored by local consumers because of its unique flavor and food quality [1]. Generally, freshwater fish (grass carp and carp) are the main raw materials used for Suanyu production. The preparation of traditional Suanyu mainly involves mixing salted freshwater fish with a certain amount of cooked corn flour and sealing for a certain period. 

When the product is mature, it will produce a unique flavor and have a long shelf life. However, the traditional fermented fish represented by sour fish are easily polluted in the process of processing, mainly in small workshops. For example, it is fermented and eaten several times a year, which will lead to a decrease in the quality and safety of the product. In addition, because the microorganisms used for the production of traditional Suanyu originate mainly from the fish itself or the environment, the fermentation cycle is long and the fermentation conditions are difficult to control, which will inevitably lead to the breeding of harmful microorganisms. Therefore, it is necessary to conduct in-depth research on the development of inoculation fermentation methods to improve the nutrition, safety, and flavor quality of fermented products and to promote the development of fermented products production industry. In the past, pure culture fermentation was commonly used. Inoculation fermentation is a new type of fish food processing method. Due to the specific enzymes and microorganisms selected, the contamination of miscellaneous bacteria was effectively limited by inoculated fermentation, and the stability and safety of the fermentation process was ensured by inoculated fermentation [2]. At the same time, the fermentation time was shortened and the flavor of the product was improved. However, few enzymes and microorganisms are involved, and few metabolites were produced, which may result in a monotonous flavor or sensory quality of the final fermented food [3].

In recent years, mixed culture fermentation with two or more starter cultures has been widely used in various fields of fermentation, including fish fillet [4], fish sauce [5], and fish intestine [6]. The synergistic effects of various starter cultures and the interactions between enzymes and microorganisms are believed to speed up the fermentation process and improve the quality and flavor of fermented products [7]. Studies have reported that, compared with single-culture fermentation, the mixed culture fermentation strains can enhance competitiveness so that the growth of harmful microbes is suppressed, guaranteeing the quality and safety of fermentation products [4,8]. In the application of meat products, mixed fermentation studies mostly the synergistic promotion between bacteria and bacteria, such as lactic acid bacteria (LAB) combined with lactic acid bacteria fermentation [9] or LAB combined with staphylococcus fermentation [10], while fungi and bacteria are rarely put together for joint fermentation.

LAB has been widely used as a starter culture, which can reduce the pH of fish [11], inhibit the growth of spoilage microorganisms, and promote the decomposition of proteins under the action of its metabolic enzymes, thus producing some unique flavors. In addition, some lactic acid bacteria have been proved to have biological activities as probiotics [12]. However, it has been found that *Lactobacillus* alone, thought to be primarily responsible for acidification to ensure product safety, are insufficient to improve the overall flavor of fermented products [13]. Yeast, a typical food probiotic not yet widely used in industrial fish, can add color and flavor to products because glucose is broken down under anaerobic conditions into alcohol and carbon dioxide, and alcohol interacts with fatty acids to form esters [14]. Yoshikawa et al. [15] found that some of the substances responsible for the soy sauce flavor of fermented salmon were produced by microbial metabolism along with a significant increase in free amino acids and ethanol. The effects of single LAB fermentation and yeast fermentation on fish meat products have been widely known [16,17], while combined fermentation of the two on fish meat products are rarely reported. It was found that the content of nitrogen compounds in shark NUKA-AES was reduced by *Saccharomycetes* and *Lactobacillus* combined fermentation, resulting in good flavor of the product [18]. In addition, acidification of the sample matrix caused by lactic acid bacteria may favor the growth of yeast, which, in turn, may release vitamins and other growth factors that stimulate LAB growth [19,20]. 

Therefore, in this study, *L. plantarum* and *S. cerevisiae* were selected as starter cultures to explore their effects on the physical and chemical characteristics and flavor of sour fish during fermentation and maturation, hoping to make contributions to shortening the fermentation cycle of sour fish. This study can provide helpful information for improving the utilization value and fermentation quality of Suanyu.

## 2. Materials and Methods

### 2.1. Preparation of Starter Culture

First, the lyophilized powder was activated in MRS broth and stored in a slant at 4 °C. Single colonies of *L. plantarum* (Beijing Beina Chuanglian Biotechnology Research Institute No. 192567, 8026, 8/F, No. 102, Dongsanhuan South Road, Chaoyang District, Beijing) were isolated in MRS Broth (Beijing Luqiao Technology Co., Ltd., Beijing, China). Then, 0.1 mL of bacterial suspension was inoculated into 10 mL MRS broth and incubated at 37 °C for 16 h. A single colony of *S. cerevisiae* (No. 336054, Beijing Beina Chuanglian Biotechnology Research Institute) was selected in YM Broth (Beijing Luqiao Technology Co., Ltd., Beijing, China). Then, 0.1 mL of bacterial suspension was inoculated into 10 mL YM broth and incubated at 30 °C for 14 h. After culture, the bacterial solution was centrifuged at 4 °C and 9700 r/min for 10 min, the supernatant was removed, and the bacterial cells were collected, washed twice with an equal volume of normal saline, then resuspended in normal saline. The final concentration was adjusted to 108~109 cfu/mL.

### 2.2. Preparation of Fermented Samples

Fresh grass carp (2500 g~3000 g) was purchased from Lingang Industrial Supermarket in Pudong New Area, Shanghai. After the live grass carp was stunned by a blow to the head, it was decapitated, the viscera removed, and the back meat was rinsed and cut into rectangular pieces (3 cm × 3 cm × 2 cm). The Suanyu samples were prepared according to the methods of Xu et al. [2], with some slight modifications. The back meat, white sugar (2%, *w*/*w*), and salt (3%, *w*/*w*) were mixed, and the mixture was evenly stirred and pickled at 4 °C for 48 h. Salted fish, fried cornmeal (25%, *w*/*w*), and bacterial solution (1%, *w*/*w*) were mixed well. The mixture was loaded into a fermenter, sealed with water, and fermented in an incubator at 30 °C for 15 days. Samples were taken at 0, 5, 10, and 15 days for further testing and stored at −80 °C. A total of 3 groups of fermented samples were designed: NF group (natural fermentation without bacteria), LP group (*L. plantarum* inoculated), and MF group (*L. plantarum*, *S. cerevisiae* inoculated (1:1 by volume, 1% inoculation)).

### 2.3. Determination of pH and Water Activity (Aw)

The pH was measured using a digital pH meter (Mettler Toledo FE28, Shanghai, China) according to the method of Hua [3]. Aw values were determined by a water activity meter (Decagon Devices, Pullman, WA, USA) at room temperature.

### 2.4. Determination of Thiobarbituric Acid Reactive Substance (TBARS)

TBA was determined according to Tian’s method with a slight modification [21].

### 2.5. Determination of Total Volatile Base Nitrogen Content (TVB-N)

The value of TVB-N was measured by FOSS 8400 Kjeldahl using Wang’s method [22].

### 2.6. Microbial Analysis

#### 2.6.1. Total Viable Counts

The determination of bacterial counts was based on Xu’s [23] method with slight modifications. The sample (2 g) and 18 mL of sterile physiological saline were added to a sterile bag, and after aseptic homogenization for 3 min, the mixture was diluted 10 times with sterile physiological saline. The sample solution (1 mL) was taken and poured into different culture media to determine the number of microorganisms. The total aerobic bacteria were cultured on plate count agar (Beijing Land Bridge Technology Co., Ltd., Beijing, China) at 30 °C for 72 h. LAB was cultured in MRS medium at 37 °C for (72 ± 2) h. Yeast counts were cultured in YPD medium at 30 °C for 48 h, 2 replicates per sample, and the number of colonies determined is expressed as lg (CFU/g).

#### 2.6.2. High-Throughput Sequencing and Sequence Analysis

After total DNA from fish samples was extracted, universal primers were designed to amplify the V4 fragment of the bacterial 16 sDNA sequence. The forward primer is 341F (CCTAYGGGRBGCASCAG), and the reverse primer is 806R (GGACTACNNGGGTATCTAAT). To detect PCR-amplified products, 1.0% agarose gel electrophoresis was used. The target fragments of the product were recovered and sequenced on an Illumina MiSeq PE250 (Illumina MiSeq, CA, USA). Lima (v1.7.0, Illumina Inc.) software was used to perform quality control on each sample sequence to obtain the Raw-sequence. The cutadapt (1.9.1, Illumina Inc.) software was used to identify and remove primer sequences and perform length filtering to obtain Clean-sequences that do not contain primer sequences. UCHIME (v4.2, Illumina Inc.) software was used to identify and remove chimeric sequences to obtain Effective-sequences. Usearch (Illumina Inc.) software was used to cluster Reads at a similarity level of 97.0% and obtain OUT. QIIME2 (Illumina Inc.) software was used to evaluate the Alpha Diversity Index of samples, including Ace, Chao1, Shannon, and Simpson (Chao1 and Ace indices were used to calculate the number of bacterial species, and Shannon and Simpson indices were used to estimate bacterial diversity).

### 2.7. Determination of Free Amino Acids

FAAs in fermented samples were determined by the method of Shi with some modifications [24]. Samples (2 g) were added to 15 mL of cold trichloroacetic acid solution, and homogenized by fm-200 homogenizer (Shanghai Fokker Equipment Co. Ltd., Shanghai, China) for 1 min, left for 2 h, then centrifuged at 10,000 r/min for 15 min at 4 °C. After filtration, the supernatant (5 mL) was taken, the pH was adjusted to 2.0 with NaOH (3 M), and the volume was quickly fixed to 10 mL with ultrapure water. After passing through a 0.22 μm filter membrane, the mixture was used for testing. Free amino acids were determined and analyzed by an automatic amino acid analyzer (L-8800; Hitachi, Tokyo, Japan).

### 2.8. Determination of Volatile Compounds

The volatile compounds were picked up by fibers surfaced with Carboxen/Dimethicone (50 μm, 1 cm) and separated and analyzed using GC-MS on an Agilent system. Xiao’s method was adopted to measure the changes of volatile compounds in the fermentation of Suanyu [25]. The sample (5 g) was placed in a 20 mL sealed headspace vial, and 2,4,6-pyridine solution was added as an internal standard. The headspace vial was agitated in a water bath at 60 °C for 10 min, and then SPME fibers adsorbed volatile substances in the headspace of the headspace vial for 40 min. Immediately after the adsorption, the SPME fiber was inserted into the GC-MS injector and desorbed by TDU at 250 °C for 5 min. Volatile compounds were analyzed with a DB-5MS capillary column (60 mm in length, 0.32 mm in inner diameter, and 1 μm in film thickness). The injections were performed in split mode. The carrier gas was helium, and the flow rate was 1.2 mL/min. The heating program was as follows: 40 °C started at a rate of 5 °C/min to 100 °C, the initial temperature was held for 1 min then increased to 180 °C at a rate of 3 °C/min, finally increased to 240 °C with a rate of 5 °C/min, then final hold for 5 min. MS conditions were as follows: ion source (ionization energy, 70 eV; emission current, 200 μA; ion source temperature, 230 °C; quadrupole temperature, 150 °C) and detector (detector voltage, 1200 V; detector temperature, 250 °C; detector interface temperature, 280 °C; mass range, 35–450 amu). Identification of volatile compounds in samples was conducted by matching MS library (Wiley/NIST 2008) and linearity retention index (LRI), while quantitative analysis of volatile compounds was carried out with internal standard method.

### 2.9. Statistical Analysis

All tests were performed in triplicate, and data were expressed as the means ± SD. Statistical analysis was performed using SPSS 23.0 software (IBM, Armonk, NY, USA), and significance (*p* < 0.05) was analyzed by one-way analysis of variance and Duncan’s multiple range test.

## 3. Results and Discussion

### 3.1. pH and Aw

pH value is an important index used to evaluate food safety. It is generally believed that the formation of bad bacteria can be inhibited by low pH, to ensure the safety of food. As presented in Figure 1A, the pH of each group of Suanyu sample showed a downward trend throughout the fermentation process. The pH of the samples when unfermented was 6.74. In the first half of fermentation (from day 0 to day 5), the pH of the three groups of samples decreased rapidly, which may be related to the decomposition of carbohydrates to produce organic acids [26]. In the second half of fermentation (from day 5 to day 15), the pH decreased gently in the three groups of samples. At the end of fermentation, the pH values of the NF, LP, and MF groups were 5.43, 4.41, and 4.35, respectively, indicating that the inoculation of lactic acid bacteria could promote acid production [1]. However, the pH difference between the LP and MF groups did not change significantly (*p* < 0.05), indicating that the addition of *S. cerevisiae* had little effect on the pH during fermentation.

The changes of Aw in Suanyu samples during the fermentation were shown in Figure 1B. The Aw value of fresh fish is 0.95. Aw values of all fermentation samples decreased during fermentation. Aw in the inoculated group was consistently significantly lower than in the natural fermentation group (*p* < 0.05), which may be related to the decrease in pH during the fermentation process. Studies have shown that the reduction of pH value after lactic acid bacteria fermentation promotes the activity of endogenous protease, which further promotes the hydrolysis of protein and increases the content of water-soluble protein, thus reducing the water holding capacity of the product, and thus reducing AW [27]. After fermentation, the final AW value was between 0.90 ± 0.92. Meanwhile, the Aw value of MF group and LP group was 0.90, and inoculation of *S. cerevisiae* had no significant effect on Aw of fermented Suanyu samples (*p* < 0.05).

### 3.2. TBA

The degree of fat oxidation in the product can be reflected by the TBA value [28]. The changes of TBARS value of the three groups of Suanyu samples during fermentation are shown in Figure 1C. The TBARS values of each group samples showed a trend of increasing first and then decreasing during fermentation. The TBARS value of fresh fish was 0.034 ± 0.005 mg/kg, and the TBARS value of NF group samples reached the maximum value of 0.110 ± 0.006 mg/kg after 10 days of fermentation. Similarly, the TBARS value of LP and MF group samples also reached maximum values on the 10th day of fermentation, which was 0.149 ± 0.004 mg/kg and 0.124 ± 0.004 mg/kg, respectively. 

The TBARS value of the inoculated group was significantly higher than that of the non-inoculated group (*p* < 0.05), which was consistent with the results of Li [29]. Some studies have shown that the addition of *L. plantarum* may participate in the process of degrading hydroperoxides to malondialdehyde at the early stage of fermentation, thus promoting the oxidation of fat [2]. In addition, at the end of fermentation, a significantly higher TBARS value existed in the NF group sample, followed by the LP and MF group samples (*p* < 0.05), indicating that the combination fermentation with *L. plantarum* and *S. cerevisiae* could effectively inhibit the oxidation of lipids and improve the antioxidant capacity of the products. Similar results were obtained by Chen et al. [10].

### 3.3. TVB-N

TVB-N is a commonly used method to judge the freshness of fish products. There is a certain correlation between the content of TVB-N and the content of nitrogenous compounds produced in the process of corruption. Higher values of TVB-N indicate higher levels of corruption. It can be seen from Figure 1D that the content of TVB-N in the samples at the end of fermentation is 21.83~26.31 mg/100 g, which is lower than 35 mg/100 g, indicating that the safety quality of the products in this fermentation meets the requirements. With the increase of fermentation time, the TVB-N values of all samples increased. The TVB-N values of the inoculated group were always significantly lower than those of the uninoculated group, which indicated that the addition of the starter could effectively inhibit the breeding of spoilage microorganisms in the products. It has been reported that increased acid neutralizes basic ammonia and amines in samples inoculated with *L. plantarum*, thereby reducing the content of TVB-N [30]. In addition, at the end of fermentation, the TVB-N values of LP and MF groups were 23.32 mg/100 g and 21.83 mg/100 g, respectively, and the TVB-N values of the mixed inoculated group were significantly lower than those of the single inoculated group (*p* < 0.05). Similar results were obtained by Hu [31], which indicated that the propagation of spoilage microorganisms could be inhibited by *L. plantarum* combined with *S. cerevisiae* fermentation; thus, the safety of the product has also been enhanced.

### 3.4. Analysis of Microorganism

#### 3.4.1. Microbial Community Analysis

The change of microbial community of the three groups of samples during fermentation are shown in Figure 2. The number of colonies in the inoculated group was significantly higher than that in the uninoculated group, indicating that inoculating the starter culture affected the number of microorganisms (Figure 2A). As shown in Figure 2B, in the first 5 days of fermentation, the number of lactic acid bacteria in the three groups increased rapidly and reached the maximum values, which were 8.295 ± 0.055 (NF), 8.8 ± 0.04 (MF), and 8.89 ± 0.19 (LP), in descending order. Then, the number of *Lactobacillus* began to decline. At the end of fermentation, the *Lactobacillus* value of LP group was 8.64 ± 0.05, which was significantly higher than 8.24 of MF group. The change of yeast counts is shown in Figure 2C. At the beginning of fermentation, the number of yeast in each group increased rapidly, among which the LP group reached the maximum on the 5th day (7.65 ± 0.06), while the NF group and MF group reached their maxima on the 10th day (7.85 ± 0.06 and 8.38 ± 0.04, respectively). After the 5th day of fermentation, the yeast counts of LP group began to decrease, finally decreasing to 7.31 ± 0.00, while the yeast counts of NF group and MF group began to decrease 10 days later, finally decreasing to 7.43 ± 0.03 and 8.11 ± 0.08. Since yeast can thrive in an acidic environment [32], yeast can continue to grow up to day 10 in an acid-producing environment caused by *Lactobacillus*. However, at the later stage of fermentation, yeast tends to show a downward trend, which is consistent with the results obtained by Halm et al. [33].

#### 3.4.2. Bacterial Diversity Analysis

Alpha diversity index was used to analyze the species abundance and diversity of individual samples. After barcode identification, 322,724 CCS (Circular Consensus Sequencing) sequences were obtained from 13 samples, and at least 16,740 CCS sequences were generated from each sample. The OTUs represents the number of species that can be detected in a single sample. As shown in Table 1, the coverage rates of the samples in this study are all above 0.99, indicating that the sample detection had a good accuracy. The Chao1 and Ace indices of NF and MF groups decreased with increasing fermentation time, indicating that species richness decreased with fermentation. As the fermentation progressed, each group of Shannon index and Simpson index gradually reduced, suggesting that the diversity of the species gradually reduced. The Shannon index and Simpson index of NF group were higher than those of the MF group, indicating that the combined fermentation reduced the diversity of species.

On the basis of the results of feature analysis, a high-throughput sequencing method was used to measure the changes of microbial community succession in the three groups of Suanyu samples during fermentation at two taxonomic levels (phylum and genus levels). As shown in Figure 3, *Firmicutes* and *Proteobacteria* were the main phyla in NF group samples, accounting for more than 82% of the total phyla. As the fermentation time increased, the original proportion of *Firmicutes* gradually decreased, while *Proteobacteria* gradually increased and finally reached 88.2%. Firmicutes were dominant phyla in the MF and LP group, and the content increased as the fermentation processed, the final figures reaching 98.8% and 98.6%, respectively. At the genus level, because of the addition of *L. plantarum*, *Lactiplantibacillus* occupied the main position in the MF group and the LP group, and it showed an upward trend throughout the process. The final proportion of *Lactiplantibacillus* was 97.95% and 98.10%. This is consistent with previous findings by Matti et al. [34]. *Enterobacter* is one of the most common pathogens in fish and its products, which usually comes from the fish itself. In the natural fermentation group, *Enterobacteria* accounted for 63.31% in the late fermentation period. Compared with the inoculation group, the number of *Enterobacteria* gradually decreased. However, at the 15th day of fermentation, *Enterobacteria* was not detected in the MF group, which was due to the inhibition of *L. plantarum*. In the MF group, with the progress of fermentation, other microorganisms related to spoilage, such as *Staphylococcus* aureus (2.91%) and *Micrococcus* (3.1%) in the early stage, could not be detected in the final stage. Only a small number of *Bacillus* and *Streptococcus* were left, less than 1%. The abundance of bacterial species was lower in the inoculated group than in the uninoculated group, likely because *L. plantarum* resisted other bacteria and was more competitive for nutrients.

### 3.5. FAAs

FAA (free amino acid) is an important factor affecting the flavor of fermented fish products, which can produce a variety of taste characteristics, including umami, sweet, and bitter. Umami amino acids include Asp and Glu; sweet amino acids include Thr, Ser, Gly, Ala, Val, and Lys; and bitter amino acids include Cys, Met, Ile, Leu, Tyr, Phe, and His. Table 2 records the content of amino acids in Suanyu samples of different fermentation methods at 15 days. The highest concentration of free amino acids (1096.73 mg/100 g) was found in the MF group samples, followed by the LP group samples (1065.52 mg/100 g) and the NF group samples (1017.67 mg/100 g). The content of the NF group samples increased from 526.26 mg/100 g to 1017.67 mg/100 g at 0 d, which confirmed the critical role of endogenous enzymes in proteolysis [35].

In addition, the contents of the MF group and LP group at the end of fermentation were significantly higher than those of the non-inoculated group, which may be attributed to the proteolytic effect of microbial enzymes released by *L. plantarum*. The amino acid concentration of MF group was higher than that of LP group, so the advantage of mixed inoculation was more pronounced. Among the amino acids detected, the concentration of His is the highest, which can promote the product to produce the characteristic seafood flavor [36]. However, histidine content should not be too high, otherwise a bitter taste will be produced, and the flavor of the fish will be affected [37]. In addition, Gly, Lys, Ile, Cys, Ala, Glu, and Phe were high, with Glu, Ala, Ile, and Lys, exceeding their thresholds and contributing significantly to the taste of fermented fish, while some amino acids, such as Val, Met, and Pro, were well below the thresholds and had little effect on flavor. At the end of fermentation, both umami and sweet amino acids were significantly increased in the inoculated group compared with the uninoculated group. In the LP and MF groups (*p* < 0.05), umami amino acids increased by 10.14% and 20.56%, sweet amino acids increased by 17.51% and 34.14%, and bitter amino acids decreased by 2.92% and 8.23%, respectively, which proved that inoculation fermentation, especially mixed fermentation, had a more prominent effect on protein hydrolysis, promoted the decomposition of amino acids, and produced beneficial flavor for the products. Endogenous and microbial enzymes exert synergistic and additive effects, thereby promoting FAA release, as demonstrated in the study of Chadong et al. [38]. In this study, by inoculation of *L. plantarum* and *S. cerevisiae*, the effects of endogenous proteases and microbial enzymes were enhanced, the decomposition of proteins was promoted, and the maturation of fish was accelerated. At the same time, the competitiveness of microorganisms with proteolytic activity was strengthened, and the generation of bitter taste was inhibited, thus promoting the production of umami and sweet flavors. Xiao et al. [26] found that adding *L. plantarum* and *Streptococcus* xylose could promote the generation of FAA and further improve the flavor of sausage, which is consistent with the results of the present study.

### 3.6. Volatile Compounds

Volatile compounds were detected from all the fermented Suanyu samples, including 13 alcohols, 11 aldehydes, 4 ketones, 8 esters, 3 aromatic compounds, 4 acids, 8 hydrocarbons, and 3 other compounds. Meanwhile, 20, 37, and 47 compounds were detected in the NF, LP and MF group fermented samples, respectively (Table 3). The flavor substances obtained by fermentation in MF group were significantly higher than those in the other two groups, which indicated that the combined inoculation of *L. plantarum* and *S. cerevisiae* has an advantage in the production of volatile compounds during Suanyu fermentation. In addition, among the three groups of fermented samples, the highest content of volatile compounds was presented in the MF group samples, followed by the LP and NF group samples, among which ketones, alcohols, and aromatic compounds had the highest concentration.

Alcohols are the most abundant compounds at the end of fermentation in the MF group, which may be due to the inoculation of *Saccharomyces cerevisiae*. Amino acids are converted by *S. cerevisiae* to their respective heteroalcohols through the Ehrlich pathway, and amino acids are transaminated, followed by decarboxylation and dehydrogenation, to form corresponding alcohols [39]. Among them, the content of Phenylethanol is the highest, which has a sweet rose-like fragrance. 1-octene-3-alcohol is the main substance produced by unsaturated fatty acids, which has a mushroom-like aroma [40]. It is also considered one of the primary alcohol flavor substances of fermented fish products [41]. In addition, the contents of n-hexanol (fruit flavor), n-octanol (wine flavor), and 3-methyl-1-butanol (grass flavor and wine flavor) in the inoculation fermentation group were also significantly higher than those of the natural fermentation group. The changes in these alcohols played a vital role in the formation of the unique flavor of fermented grass carp.

Aldehydes are usually produced due to the oxidation and degradation of fats, which generally exhibit a grassy fatty aroma and can strongly influence the flavor due to their low threshold. In this study, hexanal (grassy and fishy flavor), nonanal (floral and grassy fragrance), and heptanal (citrus aroma) are all straight-chain aldehydes, which are formed from unsaturated fatty acids [42]. Among them in the MF group, the content of nonanal is the highest, followed by hexanal and heptanal, which combined to improve the overall flavor in the fermentation process. Benzaldehyde (mushroom and almond flavors) was detected only in the LP and MF groups, suggesting that the volatile compound was produced by *L. plantarum* catabolism, which is consistent with the results of Chen et al. [43].

Four ketones were detected in all samples, among which 3-octanone was the key ketone aroma compound in the fermentation process. 3-octanone has a mushroom flavor, cheese flavor, and mildew flavor [44,45]. Moreover, the threshold of 3-octanone is low, which is beneficial to the fermentation flavor generation. The content of 3-octanone in MF group was higher than that in the LP and NF groups, which indicated that the addition of *S. cerevisiae* contributed to the production of 3-octanone.

Esters are essential aromatic compounds in fermented fish products. In the presence of micro-organisms, acids and alcohols have a high glycolytic or lipolytic activity and react with each other to form lipids, which give a fruity flavor. In this study, only two and three esters were detected in the NF group and LP group, respectively. However, nine esters were detected in the LP group, and most of those detected were ethyl esters. Ethyl ester is a type of ethanol and acid condensation product. It is worth mentioning that the number of ethyl esters detected in the MF group was significantly higher than in the other two groups (NF and LP group), which may be due to the reaction between ethanol produced by Saccharomycete fermentation and acid compounds. In this study, ethyl caproate, ethyl octanoate, and 3-methyl-1-butanol acetate are the main esters in the fermentation process. Among them, ethyl caproate has an intense fruity and wine aroma, and ethyl caprylate has a brandy aroma. These esters all contribute to the formation of fermented flavors.

Phenolic compounds are hydroxy-containing derivatives of aromatic hydrocarbons. In this study, only two phenolic compounds (eugenol and 3-allyl-6-methoxyphenol) were detected from all samples, of which eugenol was the most abundant volatile compound during fermentation, with a pleasant and clove-like odor, playing a vital role in fermented flavor development. Eugenol, the main component of most seed leaf essential oils [46], has also been shown to promote aromatic flavor formation in fermented sour fish [47]. The hydrocarbons were divided into alkanes and terpenes, and alkanes were considered to contribute little to the flavor due to their higher threshold [41]. Caryophyllene and humulene are terpenoid flavor compounds produced during fermentation, with a clove-like aroma. In addition, four acid compounds were detected in the early stage of fermentation, including tetradecanoic acid, n-Hexadecanoic acid, octadecanoic acid, and oleic acid. Since the threshold of long-chain fatty acids is small, they contribute little to the flavor, and their main role is to participate in the formation of esters in the late fermentation stage.

## 4. Conclusions

The effects of *L. plantarum* and *S. cerevisiae* inoculation on the physicochemical characteristics and flavor of Suanyu during fermentation were studied, with the natural fermentation group and *L. plantarum* fermentation group as controls. The results showed that the pH, aw, TVB-N, and TBA values of the MF group were significantly lower than those of the natural fermentation group, which indicated that the undesirable microorganisms were effectively inhibited, and the anti-fat oxidation capacity of the product was also improved by inoculation with *L. plantarum* and *S. cerevisiae*. During the fermentation process, the bacterial diversity of the MF group decreased due to the inoculation of *L. plantarum*. LAB, as the dominant bacteria, can effectively limit the growth of spoilage bacteria, which could improve the flavor of fermented products. At the completion of fermentation, the highest content of umami and sweet amino acids and volatile compounds (especially alcohols and esters) were found in the MF group samples, followed by the NF and LP group samples, indicating that combined fermentation could give Suanyu a better flavor quality. Overall, combined fermentation with *L. plantarum* and *S. cerevisiae* improved the quality and flavor characteristics of the product. This study provides a specific theoretical basis for the development of higher quality freshwater fish fermented fish products and industrialization.

## Figures and Tables

**Figure 1 foods-11-04085-f001:**
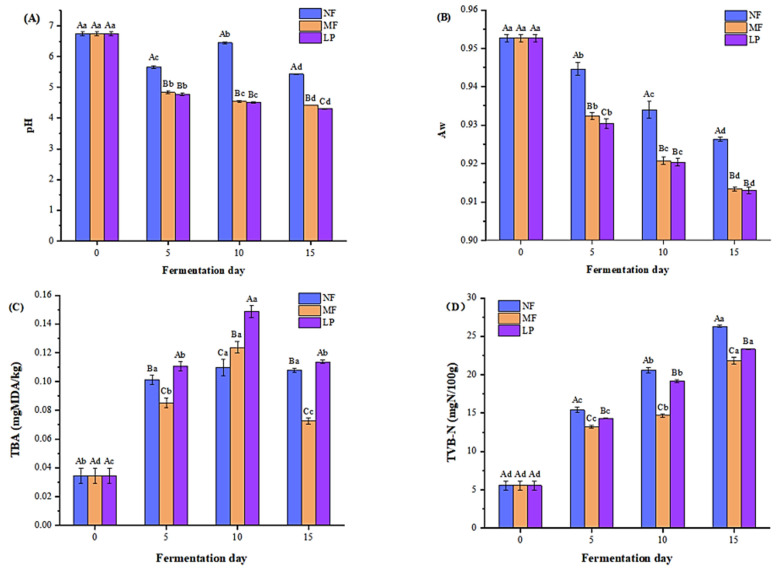
Changes in the pH (**A**), water activity (**B**), thiobarbituric acid reactive substance (**C**), and total volatile base nitrogen (**D**) of Suanyu samples with different fermentation conditions during fermentation. NF, natural fermentation; LP, inoculated *L. plantarum*; MF, inoculated *L. plantarum* combined with *S. cerevisiae*. Different letters on the bar indicate significant differences (*p* < 0.05).

**Figure 2 foods-11-04085-f002:**
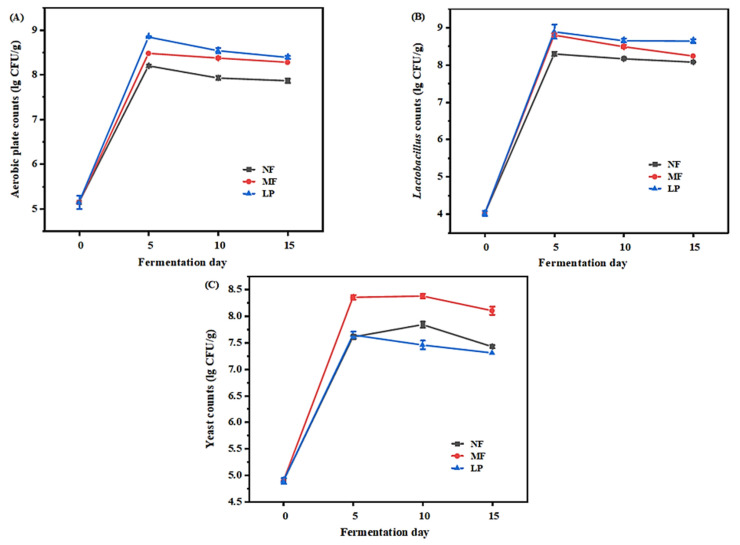
Changes in the aerobic plate counts (**A**), *Lactobacillus* counts (**B**), and yeast counts (**C**) of Suanyu samples with different fermentation conditions during fermentation. NF, natural fermentation; LP, inoculated *L. plantarum*; MF, inoculated *L. plantarum* combined with *S. cerevisiae*.

**Figure 3 foods-11-04085-f003:**
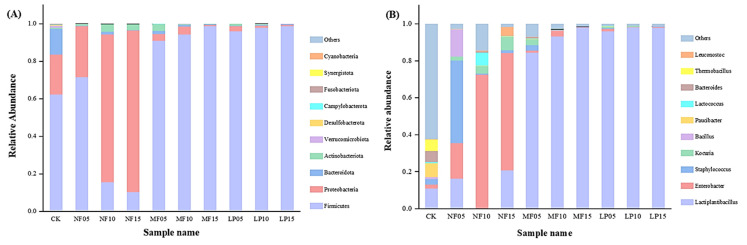
Changes in relative abundance of the microbial community of different Suanyu samples at the phylum level (**A**) and at the genus level (**B**). CK, 0 day fermentation samples; NF, natural fermentation; LP, inoculated *L. plantarum*; MF, inoculated *L. plantarum* combined with *S. cerevisiae*.

**Table 1 foods-11-04085-t001:** Alpha diversity index statistics of different fermented Suanyu samples.

Sample	Fermentation Days	OTUs	ACE	CHAO1	Simpson	Shannon	Coverage
CK	0	208.50 ± 4.50	214.85 ± 15.53	212.26 ± 14.99	0.97 ± 0.00	6.26 ± 0.23	1
NF	5	51.00 ± 9.00	75.24 ± 9.77	63.29 ± 10.04	0.72 ± 0.00	2.23 ± 0.05	1
	15	44.00 ± 4.00	99.07 ± 11.50	55.58 ± 7.72	0.46 ± 0.01	1.34 ± 0.34	1
LP	5	70.00 ± 4.00	135.47 ± 6.54	98.71 ± 10.29	0.11 ± 0.03	0.55 ± 0.08	1
	15	90.00 ± 2.00	170.94 ± 24.83	125.01 ± 15.13	0.09 ± 0.04	0.32 ± 0.06	1
MF	5	88.50 ± 3.50	151.87 ± 11.89	176.8 ± 16.60	0.20 ± 0.04	0.58 ± 0.05	1
	15	83.00 ± 0.00	135.59 ± 12.54	122.66 ± 10.13	0.04 ± 0.00	0.25 ± 0.01	1

Note: CK, 0 day fermentation samples; Coverage, sample library coverage; NF, natural fermentation; LP, inoculated *L. plantarum*; MF, inoculated *L. plantarum* combined with *S. cerevisiae*.

**Table 2 foods-11-04085-t002:** The contents of free amino acids in different fermented Suanyu samples at 15 days (mg/100 g).

FAA (mg/100 g)	Taste Characteristics	Threshold (mg/100 g)	CK	NF	LP	MF
Asp	Umami (+)	100	4.19 ± 0.23 ^d^	49.07 ± 0.67 ^c^	57.58 ± 0.87 ^b^	60.05 ± 0.13 ^a^
Thr	Sweet (+)	260	7.26 ± 0.3 ^b^	12.15 ± 0.45 ^a^	6.77 ± 0.21 ^b^	12.10 ± 0.17 ^a^
Ser	Sweet (+)	150	6.86 ± 0.22 ^c^	20.65 ± 0.85 ^a^	16.21 ± 0.71 ^b^	21.04 ± 0.78 ^a^
Glu	Umami (+)	30	2.21 ± 0.05 ^d^	71.33 ± 1.64 ^c^	75.03 ± 1.32 ^b^	85.11 ± 0.87 ^a^
Gly	Sweet/umami (+)	130	90.14 ± 4.08 ^d^	103.60 ± 2.36 ^c^	111.04 ± 1.09 ^b^	115.67 ± 6.50 ^a^
Ala	Sweet/umami (+)	60	35.43 ± 1.67 ^d^	72.69 ± 1.86 ^c^	79.85 ± 1.39 ^b^	85.68 ± 1.10 ^a^
Cys	Bitter/sweet/ surfur (−)	-	6.52 ± 2.59 ^d^	106.19 ± 2.77 ^a^	96.65 ± 0.21 ^b^	86.80 ± 1.65 ^c^
Val	Sweet/bitter (−)	40	4.50 ± 0.31 ^b^	7.18 ± 0.05 ^a^	3.84 ± 0.10 ^c^	4.06 ± 0.16 ^c^
Met	Bitter/sweet/ surfur (−)	30	0.04 ± 0.01 ^d^	1.22 ± 0.08 ^c^	5.55 ± 0.15 ^a^	4.43 ± 0.36 ^b^
lle	Bitter (−)	90	9.66 ± 0.03 ^d^	79.05 ± 1.95 ^c^	94.41 ± 0.63 ^a^	88.55 ± 1.18 ^b^
Leu	Bitter (−)	190	3.83 ± 0.84 ^b^	36.71 ± 0.67 ^a^	37.84 ± 1.53 ^a^	35.26 ± 0.36 ^a^
Tyr	Bitter (−)	-	2.40 ± 0.39 ^d^	7.74 ± 0.25 ^c^	19.86 ± 0.13 ^a^	18.48 ± 0.42 ^b^
Phe	Bitter (−)	90	5.76 ± 0.17 ^c^	67.02 ± 4.05 ^b^	84.89 ± 0.86 ^a^	84.59 ± 1.68 ^a^
Lys	Sweet/bitter (−)	50	17.16 ± 0.19 ^d^	49.73 ± 0.90 ^c^	80.20 ± 1.14 ^b^	94.70 ± 1.49 ^a^
His	Bitter (−)	20	308.05 ± 17.68 ^a^	295.36 ± 6.25 ^b^	238.24 ± 2.61 ^c^	227.71 ± 4.26 ^d^
Arg	Sweet/bitter (−)	50	4.34 ± 0.34 ^d^	10.61 ± 0.28 ^b^	35.09 ± 0.91 ^a^	34.52 ± 0.52 ^a^
Pro	Sweet/bitter (−)	300	17.90 ± 1.13 ^d^	25.88 ± 0.90 ^b^	22.47 ± 0.13 ^c^	37.99 ± 0.07 ^a^
(TUAA)	-	-	6.39 ± 0.24 ^d^	120.40 ± 1.84 ^c^	132.61 ± 1.20 ^b^	145.16 ± 0.76 ^a^
(TSAA)	-	-	157.60 ± 6.73 ^d^	302.49 ± 5.90 ^c^	355.47 ± 6.11 ^b^	405.75 ± 2.11 ^a^
(TBAA)	-	-	362.26 ± 18.48 ^d^	594.78 ± 15.41 ^a^	577.44 ± 3.84 ^b^	545.82 ± 4.73 ^c^
(TAA)	-	-	526.26 ± 24.77 ^d^	1017.67 ± 22.86 ^c^	1065.52 ± 11.08 ^b^	1096.73 ± 7.47 ^a^

Note: Different letters (a–d) in the same row indicate significant differences (*p* < 0.05). Abbreviations: (+), pleasant taste; (−), unpleasant taste; ND, threshold was not detected; CK, 0 day fermentation samples; NF, natural fermentation; LP, inoculated *L. plantarum*; MF, inoculated *L. plantarum* combined with *S. cerevisiae*; TAA, total free amino acid; TBAA, total bitter free amino acid; TSAA, total sweet free amino acid; TUAA, total umami free amino acid.

**Table 3 foods-11-04085-t003:** The contents of volatile compounds in different fermented Suanyu samples at 15 days (mg/100 g).

Volatile Compounds	Retention Time	RI	NF	LP	MF
Alcohols					
(R,R)-2,3-Butanediol	4.217	/	397.59 ± 34.09	ND	148.01 ± 14.99
1-Hexanol	6.473	958	163.67 ± 46.54	98.11 ± 11.33	197.37 ± 44.24
1-Octen-3-ol	10.204	941	393.10 ± 62.41	263.74 ± 57.64	917.44 ± 156.87
1-Pentanol	3.807	/	ND	183.30 ± 6.02	23.58 ± 1.46
3-Methyl-1-butanol	3.201	/	ND	205.15 ± 39.24	470.69 ± 182.02
1-Heptanol	9.901	984	ND	120.03 ± 9.57	73.43 ± 13.49
1-Octanol	13.568	1003	ND	ND	170.24 ± 11.52
Phenylethyl Alcohol	15.137	1105	ND	ND	1949.62 ± 61.60
2,4-Dimethylcyclohexanol	12.11	1005	ND	ND	105.80 ± 7.11
trans-2-Octen-1-ol	13.455	1049	ND	ND	101.58 ± 13.01
2-Methyl-1-butanol	3.247	/	ND	ND	213.85 ± 25.22
3-Octanol	10.788	961	ND	147.79 ± 2.57	155.14 ± 21.98
1-Nonanol	17.179	1102	ND	ND	135.33 ± 18.61
Aldehydes					
Hexanal	4.482	745	44.39 ± 18.26	631.52 ± 49.73	378.27 ± 38.75
Heptanal	7.424	868	ND	116.80 ± 39.50	108.01 ± 17.39
Nonanal	14.748	1042	44.57 ± 20.44	283.11 ± 90.36	585.35 ± 29.51
Decanal	18.364	1147	8.30 ± 3.23	17.28 ± 3.63	25.91 ± 4.93
Hexadecanal	35.604	1821	9.17 ± 1.51	193.48 ± 61.00	34.87 ± 4.65
2-Methylbutyraldehyde	2.588	/	ND	11.10 ± 1.61	ND
Benzaldehyde	9.449	963	ND	34.01 ± 11.29	42.54 ± 4.00
Tetradecanal	31.284	1632	ND	10.56 ± 5.42	ND
Pentadecanal-	33.767	1744	ND	22.22 ± 7.29	14.04 ± 1.00
(E)-2-Octenal	13.025	1035	7.80 ± 0.36	62.81 ± 14.49	108.65 ± 18.95
(E)-Hept-2-enal	9.346	958	ND	16.41 ± 2.83	43.23 ± 2.06
Ketones					
2-Heptanone	7.103	853	ND	33.98 ± 11.22	ND
2,5-Octanedione	10.364	947	ND	63.84 ± 5.98	ND
3-Octanone	10.445	949	90.95 ± 13.34	76.63 ± 8.06	193.56 ± 41.46
2-Nonanone	14.29	1031	ND	ND	37.40 ± 6.42
Esters					
Ethyl nonanoate	21.528	1251	9.20 ± 0.31	ND	ND
Ethyl hexadecanoate	37.842	1955	16.06 ± 0.59	17.80 ± 4.51	67.09 ± 3.45
3-Methylbutyl acetate	6.685	834	ND	29.91 ± 2.56	90.02 ± 16.53
2-Methylbutyl acetate	6.76	838	ND	ND	23.81 ± 3.97
Ethyl hexanoate	10.931	965	ND	ND	207.32 ± 26.19
Ethyl octanoate	18.106	1139	ND	ND	112.19 ± 17.81
2-Phenylethyl acetate	20.161	1196	ND	ND	49.58 ± 7.50
Ethyl tetradecanoate	35.243	1819	ND	ND	15.15 ± 2.83
Aromatic compounds					
p-Xylene	6.376	914	ND	ND	20.59 ± 3.23
Eugenol	23.559	1289	2343.18 ± 133.52	2586.28 ± 262.63	2757.55 ± 62.06
3-Allyl-6-methoxyphenol	24.497	1315	16.46 ± 2.39	18.84 ± 9.84	24.57 ± 1.73
Acids					
Tetradecanoic acid	34.838	1774	ND	ND	5.30 ± 0.17
n-Hexadecanoic acid	37.527	1922	ND	ND	115.92 ± 4.30
Octadecanoic acid	39.69	2046	ND	ND	90.30 ± 1.97
Oleic Acid	38.139	1969	ND	4.43 ± 0.29	ND
Hydrocarbons					
p-Cymene	13.941	1025	ND	ND	39.98 ± 2.00
Undecane	18.553	1158	39.66 ± 0.61	34.92 ± 6.50	73.93 ± 16.80
Dodecane	20.158	1209	ND	54.21 ± 6.82	ND
Tridecane	23.596	1330	21.28 ± 4.35	23.42 ± 2.07	41.39 ± 3.75
Heptadecane	33.43	1740	ND	ND	9.15 ± 0.79
Caryophyllene	28.424	1518	256.74 ± 36.64	76.81 ± 4.91	204.50 ± 57.17
Humulene	29.483	1563	61.64 ± 14.28	32.71 ± 5.15	55.31 ± 8.99
Caryophyllene oxide	30.368	1600	25.92 ± 2.71	24.44 ± 2.11	35.68 ± 8.42
Other compounds					
Trichloromethane	1.833	/	40.35 ± 5.33	20.83 ± 2.91	ND
Oxime-, methoxy-phenyl-_	8.436	882	115.90 ± 22.43	191.13 ± 21.98	175.18 ± 57.62
Carbamodithioic acid, diethyl-, methyl ester	24.177	1619	ND	210.63 ± 23.52	253.95 ± 27.07

Note: RT, compounds were validated by the retention index of the standard compound; ND, not detected. NF, natural fermentation; LP, inoculated *L. plantarum*; MF, inoculated *L. plantarum* combined with *S. cerevisiae*.

## Data Availability

Data is contained within the article.

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
