# Peer review of "Changes of Physicochemical Characteristics and Flavor during Suanyu Fermentation with Lactiplantibacillus plantarum and Saccharomyces cerevisiae"

_foods, 2022, doi:10.3390/foods11244085_

Round 1

Reviewer 1 Report

Dear Editors and authors,

1-The abstract of the manuscript needs to be rewritten and some numbers added for the most prominent results.

2-Bacteria names are misspelled or italicized, scientific names must be corrected throughout the manuscript.

3-The new name should be used for lactic acid bacteria, for example Lactobacillus plantarum correct to Lactiplantibacillus plantarum. 

4- The introduction needs to be supported by some important scientific references, such as: 1-Niamah, A. K. (2017). Physicochemical and microbial characteristics of yogurt with Added Saccharomyces boulardii. Current Research in Nutrition and Food Science Journal5(3), 300-307.

2-Xia, R., Wang, Z., Xu, H., Hou, Z., Li, Y., Wang, Y., ... & Xin, G. (2022). Cutting root treatment combined with low-temperature storage regimes on non-volatile and volatile compounds of Oudemansiella raphanipes. LWT166, 113754.

5-The Preparation of fermented samples method  needs to add reference.

6-Did the author use an antibiotic to prevent the growth of bacteria from the development of yeast on YM medium ?

7- Page 4, High-throughput sequencing and sequence analysis, What is the PCR program used in this method?

8- Figure 1, Figure 2, and Figure 3,  Abbreviations ( NFx, MFx, and LPx) must be clearly mentioned below the figure.

9- The culture medium does not count the numbers of Saccharomyces only, but all the numbers belonging to different types of yeasts are counted. The test must be corrected in order to calculate the numbers of yeasts and in all the manuscript.

10- The titles of the working methods (See page 3 line 134-144) in calculating the bacteria and yeast cultures do not match the titles of the graphs in Figure 2. The working methods and results must be identical.

11-Abbreviations and symbols in Table 1 and Table2 must be fully and clearly defined at the under of the table.

12-The conclusions of the manuscript contain many results and this is incorrect, it should be rewritten again.

13- The manuscript contains many spelling and grammatical errors must be re-corrected by native people , see line 144, 262,..........etc. 

Author Response

Dear Editors and Reviewers:

On behalf of my co-authors, we thank you very much for giving us an opportunity to revise our manuscript, we appreciate academic editor and reviewers very much for their positive and constructive comments and suggestions on our manuscript entitled “Changes of physicochemical characteristics and flavor during Suanyu fermentation with Lactiplantibacillus plantarum and Saccharomyces cerevisiae”. (ID: foods-2051953).

We have studied academic editor and reviewer’s comments carefully. Those comments are all valuable and very helpful for revising and improving our paper, as well as the important guiding significance to our research. Revised portions are marked in red in the paper. We have revised the content of the manuscript to reduce the repetition rate.

We tried our best to improve the manuscript and made some changes in the manuscript. These changes will not influence the content and framework of the paper. And here we did not list the changes but marked in red in revised paper.

We would like to express our great appreciation to you and reviewers for comments on our paper. Looking forward to hearing from you.

Thank you and best regards.

Yours sincerely,

Qiang Zhang

Wenzheng Shi and Weiqiang Qiu

College of Food Science & Technology, Shanghai Ocean University

No.999 Hucheng Huan Road, Shanghai, P.R. China 201306

Email: wzshi@shou.edu.cn; dhsguoqy@163.com

Dear Editors and Reviewers:

Thank you for your letter and the comments concerning our manuscript entitled “Correlation of taste components with consumer preferences and emotions in Chinese mitten crabs (Eriocheir sinensis): the use of artificial neural network model”. (ID: foods-2408995). These comments are all valuable and very helpful for revising and improving our paper, as well as the important guiding significance to our research. We have studied comments carefully and have made correction which we hope meet with approval. Revised portions are marked in red in the paper. The main corrections in the paper and the responds to the editors and reviewers are as follow:

Responds to the reviewers’ comments:

To reviewer 1

  1. Response to comment 1: The abstract of the manuscript needs to be rewritten and some numbers added for the most prominent results.

Response: Sincerely thanks for your advice, we have rewritten the abstract as you suggested and added some data.

  1. Response to comment 2: Bacteria names are misspelled or italicized, scientific names must be corrected throughout the manuscript.

Response: Sincerely thanks for your suggestions. The bacteria name has been corrected in the manuscript according to the correct format.

  1. Response to comment 3: The new name should be used for lactic acid bacteria, for example Lactobacillus plantarum correct to Lactiplantibacillus plantarum.

Response: Sincerely thanks for your suggestions. The name of the lactic acid bacteria has been corrected according to your suggestion.

  1. Response to comment 4: The introduction needs to be supported by some important scientific references, such as: 1-Niamah, A. K. (2017). Physicochemical and microbial characteristics of yogurt with Added Saccharomyces boulardii. Current Research in Nutrition and Food Science Journal, 5(3), 300-307. 2-Xia, R., Wang, Z., Xu, H., Hou, Z., Li, Y., Wang, Y., ... & Xin, G. (2022). Cutting root treatment combined with low-temperature storage regimes on non-volatile and volatile compounds of Oudemansiella raphanipes. LWT, 166, 113754.‏

Response: Sincerely thanks for your suggestions. At your suggestion, these important scientific references have been inserted in the manuscript (Line 88 and Line 402).

  1. Response to comment 5: The Preparation of fermented samples method needs to add reference.

Response: Sincerely thanks for your suggestions. We have revised and added relevant reference as suggested (Line 113).

  1. Response to comment 6: Did the author use an antibiotic to prevent the growth of bacteria from the development of yeast on YM medium?

Response: Sincerely thanks for your suggestions. We did not add antibiotics to inhibit bacterial growth, as my experiments were carried out on a sterile bench, the supplies were autoclaved and the fermentation process was carried out under anaerobic conditions with a sealed lid and water seal.

  1. Response to comment 7: Page 4, High-throughput sequencing and sequence analysis, what is the PCR program used in this method?

Response: Sincerely thanks for your comments and suggestions. The PCR program used in this method as follow:

PCR reaction system(50μL): 4μL Primer Cocktail, 25μL Master Mix, 2μL DNA and 19μL ddH2O. Reaction parameters: Pre-denaturation at 94°C for 5 min; Denaturation at 94°C for 40 s, annealing at 56 °C for 50 seconds, extension at 72 °C for 50 seconds, total 30 cycles. Final extension at 72 ℃ for 5 minutes.

  1. Response to comment 8: Figure 1, Figure 2, and Figure 3, Abbreviations (NFx, MFx, and LPx) must be clearly mentioned below the figure.

Response: Sincerely thanks for your suggestions. The figures have been modified and the abbreviations were clearly mentioned below the figure.

  1. Response to comment 9: The culture medium does not count the numbers of Saccharomyces only, but all the numbers belonging to different types of yeasts are counted. The test must be corrected in order to calculate the numbers of yeasts and in all the manuscript.

Response: Sincerely thanks for your comments and suggestions. We have retested the yeast counts using selective culture YPD (Yeast extract 10 g, peptone 20 g, glucose 20 g, fixed volume to 1 L. Add 1.5% agar powder to make a fixed medium) as you suggested, redone the figure and corrected the related contents (Line 265-275).

  1. Response to comment 10: The titles of the working methods (See page 3 line 134-144) in calculating the bacteria and yeast cultures do not match the titles of the graphs in Figure 2. The working methods and results must be identical.

Response: Sincerely thanks for your comments and suggestions. We have modified the title of the working methods according to your suggestion and modified Figure 2 to make the working methods and results be identical.

  1. Response to comment 11: Abbreviations and symbols in Table 1 and Table2 must be fully and clearly defined at the under of the table.

Response: Sincerely thanks for your suggestions. Abbreviations and symbols have been fully and clearly defined at the under of the table.

  1. Response to comment 12: The conclusions of the manuscript contain many results and this is incorrect, it should be rewritten again.

Response: Sincerely thanks for your comments and suggestions. The conclusion has been adjusted and modified for refining and summarization so that it is no longer a repetition of the results and adds to the limitations of this study and the focus of future research.

  1. Response to comment 13: The manuscript contains many spelling and grammatical errors must be re-corrected by native people, see line 144, 262, ..........etc.

Response: Sincerely thanks for your comments and suggestions. The manuscript has been re-corrected so that there are no grammatical and spelling errors.

We tried our best to improve the manuscript and made some changes in the manuscript.  These changes will not influence the content and framework of the paper. And here we did not list the changes but marked in red in revised paper.

We appreciate for Editors/Reviewers’ warm work earnestly, and hope that the correction will meet with approval.

Once again, thank you very much for your comments and suggestions.

Thank you and best regards.

Qiang Zhang

Wenzheng Shi and Weiqiang Qiu

College of Food Science & Technology, Shanghai Ocean University

No.999 Hucheng Huan Road, Shanghai, P.R. China 201306

Email: wzshi@shou.edu.cn; dhsguoqy@163.com

Reviewer 2 Report

In my opinion the manuscript exhibited high quality and is interesting patch way in the fish fermentation area.

In methodology points 2.3, 2.4 and 2.5 could be connected. 

Also, it would be worth clarifying that the YM medium was used to determine the total number of fungi not just the specific strain.

The weak point is the presentation of the study. The quality of the pictures should be improved and also the presentation of the results. Please check carefully that all abbreviations are explained and tables are well signed. Some points could be combined.

The pictures should be added in better quality. 

Once again while writing the articles I noticed the old names of the strains. Nowadays there are still studies that have these names, but for a higher quality of work, they should be improved.

I also suggested a change in the style of citations and a deeper elaboration under the introduction, especially the conclusions.

Author Response

Dear Editors and Reviewers:

On behalf of my co-authors, we thank you very much for giving us an opportunity to revise our manuscript, we appreciate academic editor and reviewers very much for their positive and constructive comments and suggestions on our manuscript entitled “Changes of physicochemical characteristics and flavor during Suanyu fermentation with Lactiplantibacillus plantarum and Saccharomyces cerevisiae”. (ID: foods-2051953).

We have studied academic editor and reviewer’s comments carefully. Those comments are all valuable and very helpful for revising and improving our paper, as well as the important guiding significance to our research. Revised portions are marked in red in the paper. We have revised the content of the manuscript to reduce the repetition rate.

We tried our best to improve the manuscript and made some changes in the manuscript. These changes will not influence the content and framework of the paper. And here we did not list the changes but marked in red in revised paper.

We would like to express our great appreciation to you and reviewers for comments on our paper. Looking forward to hearing from you.

Thank you and best regards.

Yours sincerely,

Qiang Zhang

Wenzheng Shi and Weiqiang Qiu

College of Food Science & Technology, Shanghai Ocean University

No.999 Hucheng Huan Road, Shanghai, P.R. China 201306

Email: wzshi@shou.edu.cn; dhsguoqy@163.com

Dear Editors and Reviewers:

Thank you for your letter and the comments concerning our manuscript entitled “Correlation of taste components with consumer preferences and emotions in Chinese mitten crabs (Eriocheir sinensis): the use of artificial neural network model”. (ID: foods-2408995). These comments are all valuable and very helpful for revising and improving our paper, as well as the important guiding significance to our research. We have studied comments carefully and have made correction which we hope meet with approval. Revised portions are marked in red in the paper. The main corrections in the paper and the responds to the editors and reviewers are as follow:

Responds to the reviewers’ comments:

To reviewer 2

1.Response to comment 1: In my opinion the manuscript exhibited high quality and interesting patch way in the fish fermentation area.

Response: Sincerely thanks for your comments.

2.Response to comment 2: In methodology points 2.3, 2.4 and 2.5 could be connected.

Response: Sincerely thanks for your comments and suggestions.

3.Response to comment 3: Also, it would be worth clarifying that the YM medium was used to determine the total number of fungi not just the specific strain.

Response: Sincerely thanks for your comments and suggestions. We have retested the yeast counts using selective culture YPD (Yeast extract 10 g, peptone 20 g, glucose 20 g, fixed volume to 1 L. Add 1.5% agar powder to make a fixed medium) as you suggested, redone the figure and corrected the related contents (Line 265-275).

4.Response to comment 4: The weak point is the presentation of the study. The quality of the pictures should be improved and also the presentation of the results. Please check carefully that all abbreviations are explained and tables are well signed. Some points could be combined.

Response: Sincerely thanks for your comments and suggestions. Some of the figures and the presentation of the results have been modified to improve the quality. Abbreviations and symbols have been fully and clearly defined at the under of the table.

5.Response to comment 5: The pictures should be added in better quality.

Response: Sincerely thanks for your suggestions. The pictures have been added in better quality.

6.Response to comment 6: Once again while writing the articles I noticed the old names of the strains. Nowadays there are still studies that have these names, but for a higher quality of work, they should be improved.

Response: Sincerely thanks for your suggestions. The name of the lactic acid bacteria has been corrected according to your suggestion.

7.Response to comment 7: I also suggested a change in the style of citations and a deeper elaboration under the introduction, especially the conclusions.

Response: Sincerely thanks for your suggestions. Some important documents have been inserted to do a deeper elaboration under the introduction; The conclusion has been adjusted and modified for refining and summarization so that it is no longer a repetition of the results and adds to the limitations of this study and the focus of future research.

We tried our best to improve the manuscript and made some changes in the manuscript.  These changes will not influence the content and framework of the paper. And here we did not list the changes but marked in red in revised paper.

We appreciate for Editors/Reviewers’ warm work earnestly, and hope that the correction will meet with approval.

Once again, thank you very much for your comments and suggestions.

Thank you and best regards.

Qiang Zhang

Wenzheng Shi and Weiqiang Qiu

College of Food Science & Technology, Shanghai Ocean University

No.999 Hucheng Huan Road, Shanghai, P.R. China 201306

Email: wzshi@shou.edu.cn; dhsguoqy@163.com

Reviewer 3 Report

To the Authors,

in their paper (foods-2051953) titled "Changes of physicochemical characteristics and flavor during Suanyu fermentation with Lactobacillus plantarum and Saccharomyces cerevisiae", the authors Zhang et al. "investigate[d] the changes in the physicochemical characteristics and flavor of fermented Suanyu during fermentation with Lactobacillus plantarum (L. plantarum) and Saccharomyces cerevisiae (S. cerevisiae). [They found] highest content of umami and sweet amino acids and volatile compounds (especially alcohols and esters) in MF, [so] combined fermentation could give Suanyu a better flavor quality".

The paper studies a seemingly important fish/seafood product: sour fish suanyu. It complements published literature by extending the application to this food product. Oddly, from a cursory online search, previous, recently published papers on suanyu are not e.g. https://www.tandfonline.com/doi/abs/10.1080/10942912.2017.1293089 or https://doi.org/10.1016/j.foodchem.2021.129863 or https://doi.org/10.1111/jfpp.13131 or https://doi.org/10.1016/j.foodres.2022.111631 or https://doi.org/10.3390/foods11121721 or https://doi.org/10.4315/JFP-19-607. Another issue is: Why there was no SF group i.e. S. cerevisiae only inoculation? In the M&M some more info about methods is welcomed, but authors recurred to appropriate and current/up-to-date methodologies. In the Results and Discussion, section authors could have compared their findings for some of the parameters with "threshold" values published in the literature for example pH or TBARS. In the conclusions, a few sparingly actual values complementing the sentences would be nice. A number of issues require revision.

General and some specific comments:

  • Consider adding a term (e.g. fish or seafood) to Title (and Abstract) to clarify what type of food to unknowledgeable readers, e.g. "sour fish suanyu"

  • in the Abstract, complete sci. name on 1st mention then on subsequent instances just abbreviate genus, e.g. to L. plantarum

  • L. 28-29, this is an empirical study.

  • L.35, citation(s) required.

  • L.41, clarify.

  • L.45-47, is there previous research on these natural microorganisms?

  • L. 50-53, citation(s) required

  • L.66, meaning? clarify.

  • L.71, citation(s) required.

  • L.84-85, citation(s) required.

  • L.98, what powder?

  • Why no SF (S. cerevisiae inoculation)?

  • L.122, F-T as in freezing-thawing? complete. what was the purpose of these F-T cycles? were they applied to every sample?

  • L.131, which were? consider providing a few more details about the actual procedures.

  • L.133, consider providing a few more details about the actual procedures.

  • L.155-156, citation(s) required as above for other software.

  • What about the diversity indices mentioned later in the paper, e.g. Simpson index or Shannon index?

  • L.197, avoid these "subjective" terms.

  • L.206, add stats info e.g. (ANOVA, p<0.05 and Duncan test, p<0.01); same in L.210, 218 and whenever you state significance in the text.

  • Are Aw values found during and at the end of the fermentation "good", and useful in terms of food safety?

  • Fig. 1, why are the capital/upper case and lower case letters above the bars?

  • How do the values of TBARS relate to deterioration and spoilage? compare with proposed-threshold values?

  • L.232, inoculated?

  • L.240, for which seafood product?

  • L.247, citation(s) to legislation/regulation required.

  • L.257, for which seafood product?

  • Fig. 2, What do whiskers mean?

  • L.292, 294, these indices should have been presented succinctly in the M&M section.

  • L.308, Lactiplantibacillus in the legend of Fig3B?

  • L.309, for which seafood product?

  • L.329-330, consider not using abbreviations (of FAA) at 1st mention.

  • Table 2. consider annotating (in bold?) the FAA that had concentrations > indicated threshold in the table body. Taste characteristics -- citation(s) required. Threshold -- citation required; also, the threshold for what? physiological response? sensory response?

  • L.378, "certain advantages" meaning?

  • L.445-446, anti-ox. capacity not studied, should not appear in the Conclusion.

  • Table 3. "samples at 0 and 15 days", but only 1 value per Volatile compound? clarify/complete.

Also, check the PDF where I made annotations using the tools available in Adobe Reader DC.

Author Response

Dear Editors and Reviewers:

On behalf of my co-authors, we thank you very much for giving us an opportunity to revise our manuscript, we appreciate academic editor and reviewers very much for their positive and constructive comments and suggestions on our manuscript entitled “Changes of physicochemical characteristics and flavor during Suanyu fermentation with Lactiplantibacillus plantarum and Saccharomyces cerevisiae”. (ID: foods-2051953).

We have studied academic editor and reviewer’s comments carefully. Those comments are all valuable and very helpful for revising and improving our paper, as well as the important guiding significance to our research. Revised portions are marked in red in the paper. We have revised the content of the manuscript to reduce the repetition rate.

We tried our best to improve the manuscript and made some changes in the manuscript. These changes will not influence the content and framework of the paper. And here we did not list the changes but marked in red in revised paper.

We would like to express our great appreciation to you and reviewers for comments on our paper. Looking forward to hearing from you.

Thank you and best regards.

Yours sincerely,

Qiang Zhang

Wenzheng Shi and Weiqiang Qiu

College of Food Science & Technology, Shanghai Ocean University

No.999 Hucheng Huan Road, Shanghai, P.R. China 201306

Email: wzshi@shou.edu.cn; dhsguoqy@163.com

Dear Editors and Reviewers:

Thank you for your letter and the comments concerning our manuscript entitled “Correlation of taste components with consumer preferences and emotions in Chinese mitten crabs (Eriocheir sinensis): the use of artificial neural network model”. (ID: foods-2408995). These comments are all valuable and very helpful for revising and improving our paper, as well as the important guiding significance to our research. We have studied comments carefully and have made correction which we hope meet with approval. Revised portions are marked in red in the paper. The main corrections in the paper and the responds to the editors and reviewers are as follow:

Responds to the reviewers’ comments:

To reviewer 3

1.Response to comment 1: Consider adding a term (e.g. fish or seafood) to Title (and Abstract) to clarify what type of food to unknowledgeable readers, e.g. "sour fish suanyu"

Response: Sincerely thanks for your suggestions. The term was added to abstract to clarify suanyu(a Chinese fermented fish)(Line12).

2.Response to comment 2: in the Abstract, complete sci. name on 1st mention then on subsequent instances just abbreviate genus, e.g. to L. plantarum

Response: Sincerely thanks for your suggestions. The errors in the manuscript has been corrected as your suggestion.

3.Response to comment 3: L. 28-29, this is an empirical study.

Response: Sincerely thanks for your comments. Here the “combined fermentation” refers to the combination of L. plantarum and S. cerevisiae.

4.Response to comment 4: L.35, citation(s) required.

Response: Sincerely thanks for your suggestions. Relevant citation(s) has been inserted (Line35).

5.Response to comment 5: L.41, clarify.

Response: Sincerely thanks for your suggestions. The fermentation process of traditional Chinese fermented fish is influenced by many natural conditions, such as season, temperature, sunlight, humidity, etc., and there are differences in the fermented fish products produced under different conditions. In addition, there are no strict rules for each stage of the production process, which usually depends on the experience of the workers.

6.Response to comment 6: L.45-47, is there previous research on these natural microorganisms?

Response: Sincerely thanks for your comments. General studies are carried out to determine the dynamics of the flora during fermentation by analysing the microbial diversity of traditional fermentation products. We have selected one of these studies: - Zeng, X., Xia, W., Jiang, Q., & Yang, F. (2013). Chemical and microbial properties of Chinese traditional low-salt fermented whole fish product Suan yu. Food Control, 30, 590-595.

7.Response to comment 7: L. 50-53, citation(s) required

Response: Sincerely thanks for your suggestions. Relevant citation(s) has been inserted (Line53).

8.Response to comment 8: L.66, meaning? clarify.

Response: Sincerely thanks for your comments. It has been reported that compared to monoculture fermentation, mixed culture fermentation can make the dominant strain more competitive, thus inhibiting the growth of harmful microorganisms and ensuring the quality and safety of fermented products

9.Response to comment 9: L.71, citation(s) required.

Response: Sincerely thanks for your suggestions. Relevant citation(s) has been inserted (Line71).

10.Response to comment 10: L.84-85, citation(s) required.

Response: Sincerely thanks for your suggestions. Relevant citation(s) has been inserted (Line83).

11.Response to comment 11: L.98, what powder?

Response: Sincerely thanks for your comments. the strain lyophilized powder was activated in MRS broth and stored in a slant at 4°C.

12.Response to comment 12: Why no SF (S. cerevisiae inoculation)?

Response: Sincerely thanks for your comments. The focus of this study was to highlight the advantages of combined fermentation, the other groups were presented as control groups and the SC group was not included due to the large amount of data.

13.Response to comment 13: L.122, F-T as in freezing-thawing? complete. what was the purpose of these F-T cycles? were they applied to every sample?

Response: Sincerely thanks for your comments. This content error has been corrected.

14.Response to comment 14: L.131, which were? consider providing a few more details about the actual procedures.

Response: Sincerely thanks for your comments and suggestions. Due to the length of the article, this part of the methodology was abbreviated. Here are the details of procedures: First, 1 g of sample was weighed and added to 10 mL of 7.5% trichloroacetic acid (containing 0.1% of EDTA). After being homogenized for 1 min at 15,000 rpm, samples were left in the ice bath for 10 min and then filtered. To 3 mL of filtrate in test tubes, 3 mL of 0.02 mol/L 2-thiobarbituric acid in distilled water was added. The sample solution wasreacted at boiling bath for 40 min. Finally, the absorbance was measured at 532 nm aftercooling with ice water. A standard curve was built by 1,1,3,3-Tetraethoxypropane (TEP).

15.Response to comment 15: L.133, consider providing a few more details about the actual procedures.

Response: Sincerely thanks for your comments and suggestions. Due to the length of the article, this part of the methodology was abbreviated. Here are the details of procedures: The samples (5g) were minced and mixed with 2g magnesium oxide slightly in a digestive tube. Following distillation, the volatile nitrogen was recovered and titrated with 0.1 M hydrochloric acid in a 1% boric acid solution (w/v).

16.Response to comment 16: L.155-156, citation(s) required as above for other software.

Response: Sincerely thanks for your suggestions. Citation(s) has been added in manuscript.

17.Response to comment 17: What about the diversity indices mentioned later in the paper, e.g. Simpson index or Shannon index?

Response: Sincerely thanks for your comments. The Chao1 and Ace indices measure species abundance, i.e. the number of species. the Shannon and Simpson indices are used to measure species diversity and are influenced by the abundance and Community evenness of species in the sample community.

18.Response to comment 18: L.197, avoid these "subjective" terms.

Response: Sincerely thanks for your suggestions.

19.Response to comment 19: L.206, add stats info e.g. (ANOVA, p<0.05 and Duncan test, p<0.01); same in L.210, 218 and whenever you state significance in the text.

Response: Sincerely thanks for your suggestions. Stats info has been added in the manuscript (Line203,208,215,231,236,254,349).

20.Response to comment 20: Are Aw values found during and at the end of the fermentation "good", and useful in terms of food safety?

Response: Sincerely thanks for your comments. Low water activity inhibited the multiplication of most bacteria and in this study the water activity decreased significantly as fermentation progressed, indicating that co-fermentation inhibited the growth of spoilage microorganisms. At the end of fermentation, the Aw value was 0.90, which is within the food safe range.

21.Response to comment 21: Fig. 1, why are the capital/upper case and lower case letters above the bars?

Response: Sincerely thanks for your comments. Uppercase letters indicate differences between groups, lowercase letters indicate differences within groups.

22.Response to comment 22: How do the values of TBARS relate to deterioration and spoilage? compare with proposed-threshold values?

Response: Sincerely thanks for your comments. The TBA value indicates the amount of fat secondary oxidation products (final products) and is widely used to determine the degree of fat oxidation in meat and meat products. The TBA value for better quality fish should not exceed 5 mg/kg.

23.Response to comment 23: L.232, inoculated?

Response: Sincerely thanks for your suggestions. The error has been corrected (231-232).

24.Response to comment 24: L.240, for which seafood product?

Response: Sincerely thanks for your comments. Product refers to the suanyu sample used in this experiment

25.Response to comment 25: L.247, citation(s) to legislation/regulation required.

Response: Sincerely thanks for your comments. The problem has been corrected

25.Response to comment 25: L.257, for which seafood product?

Response: Sincerely thanks for your comments. Product refers to the suanyu sample used in this experiment

26.Response to comment 26: Fig. 2, What do whiskers mean?

Response: Sincerely thanks for your suggestions. The figure has been modified and the abbreviations were clearly mentioned below the figure, the pictures have been added in better quality.

27.Response to comment 27: L.292, 294, these indices should have been presented succinctly in the M&M section.

Response: Sincerely thanks for your suggestions. These indices have been presented succinctly in the M&M section as you suggested (154-155).

28.Response to comment 28: L.308, Lactiplantibacillus in the legend of Fig3B?

Response: Sincerely thanks for your comments. Error has been corrected (304-306).

29.Response to comment 29: L.309, for which seafood product?

Response: Sincerely thanks for your comments. Here the product refers to fish products (fish and its products).

30.Response to comment 30: L.329-330, consider not using abbreviations (of FAA) at 1st mention.

Response: Sincerely thanks for your suggestions. Error has been corrected (327).

31.Response to comment 31: Table 2. consider annotating (in bold?) the FAA that had concentrations > indicated threshold in the table body. Taste characteristics -- citation(s)

Response: Sincerely thanks for your suggestions.

32.Response to comment 32: L.378, "certain advantages" meaning?

Response: Sincerely thanks for your comments. Error has been corrected (377).

33.Response to comment 33: L.445-446, anti-ox. capacity not studied, should not appear in the Conclusion.

Response: Sincerely thanks for your suggestions. The Antioxidation has been corrected to the anti-fat oxidation (446).

34.Response to comment 34: Table 3. "samples at 0 and 15 days", but only 1 value per Volatile compound? clarify/complete.

Response: Sincerely thanks for your suggestions. The value of 0 days of fermentation does not exist, error has been corrected (435).

We tried our best to improve the manuscript and made some changes in the manuscript.  These changes will not influence the content and framework of the paper. And here we did not list the changes but marked in red in revised paper.

We appreciate for Editors/Reviewers’ warm work earnestly, and hope that the correction will meet with approval.

Once again, thank you very much for your comments and suggestions.

Thank you and best regards.

Qiang Zhang

Wenzheng Shi and Weiqiang Qiu

College of Food Science & Technology, Shanghai Ocean University

No.999 Hucheng Huan Road, Shanghai, P.R. China 201306

Email: wzshi@shou.edu.cn; dhsguoqy@163.com

Round 2

Reviewer 1 Report

The authors made all necessary changes to improve the manuscript.